# The Importance of the Knee Joint Meniscal Fibrocartilages as Stabilizing Weight Bearing Structures Providing Global Protection to Human Knee-Joint Tissues

**DOI:** 10.3390/cells8040324

**Published:** 2019-04-06

**Authors:** James Melrose

**Affiliations:** 1Raymond Purves Bone and Joint Research Laboratory, Kolling Institute, Northern Sydney Local Health District, St. Leonards, NSW 2065, Australia; james.melrose@sydney.edu.au; 2Graduate School of Biomedical Engineering, University of New South Wales, Sydney, NSW 2052, Australia; 3Sydney Medical School, Northern, University of Sydney, Royal North Shore Hospital, St. Leonards, NSW 2065, Australia; 4Faculty of Medicine and Health, University of Sydney, Royal North Shore Hospital, St. Leonards, NSW 2065, Australia

**Keywords:** meniscus, knee stabilization, meniscal biomarkers, mesenchymal stem cell, osteoarthritis, rheumatoid arthritis, weight bearing

## Abstract

The aim of this study was to review aspects of the pathobiology of the meniscus in health and disease and show how degeneration of the meniscus can contribute to deleterious changes in other knee joint components. The menisci, distinctive semilunar weight bearing fibrocartilages, provide knee joint stability, co-ordinating functional contributions from articular cartilage, ligaments/tendons, synovium, subchondral bone and infra-patellar fat pad during knee joint articulation. The meniscus contains metabolically active cell populations responsive to growth factors, chemokines and inflammatory cytokines such as interleukin-1 and tumour necrosis factor-alpha, resulting in the synthesis of matrix metalloproteases and A Disintegrin and Metalloprotease with ThromboSpondin type 1 repeats (ADAMTS)-4 and 5 which can degrade structural glycoproteins and proteoglycans leading to function-limiting changes in meniscal and other knee joint tissues. Such degradative changes are hall-marks of osteoarthritis (OA). No drugs are currently approved that change the natural course of OA and translate to long-term, clinically relevant benefits. For any pharmaceutical therapeutic intervention in OA to be effective, disease modifying drugs will have to be developed which actively modulate the many different cell types present in the knee to provide a global therapeutic. Many individual and combinatorial approaches are being developed to treat or replace degenerate menisci using 3D printing, bioscaffolds and hydrogel delivery systems for therapeutic drugs, growth factors and replacement progenitor cell populations recognising the central role the menisci play in knee joint health.

## 1. Introduction

Osteoarthritis is responsible for a large and rapidly increasing global disease burden that is challenging health-care systems worldwide. The need for improved and rapid processes to develop new therapies, is obvious. Disease progression in osteoathritis, is marked by increasing pain and loss of joint function, however the development of osteoarthritis is usually slow, taking many years from its first mild symptoms to severe debilitating end-stage disease. Although progression to this end-state occurs in only a minority of patients with osteoarthritis and disease in many patients remains stable for many years, each year osteoarthritis led to ~160,000 total joint replacements in England and Wales and 492,000 in the USA in 2017 [1]. These numbers are expected to increase dramatically over the next two decades with global ageing trends effecting population demographics with a greater proportion of elderly OA prone patients becoming represented in the general population. Osteoarthritis is thus projected to become one of the leading musculoskeletal conditions by 2050.

The semilunar meniscal fibrocartilaginous menisci provide joint congruity to the curved weight bearing surfaces of the articular cartilages and collectively these structures provide weight bearing properties to the human knee joint [2] (Figure 1). The menisci are also designed to withstand shear forces and to withstand multidirectional stresses which are generated in with weight bearing and the torsional forces generated during normal knee articulation [3]. Complex collagenous fibrillar arrangements in the menisci are formed into lamellar structures and circumferential and radial tie bundles to provide the required material properties to this tissue [4,5]. Type I collagen is a major fibrillar component in the meniscus while type VI collagen forms pericellular collagenous structures around strings of meniscal cells and may have biomechanical sensory roles providing cell matrix feedback cues which allow the meniscal cells to regulate meniscal composition and tissue homeostasis. Several proteoglycans have been identified in the meniscus including aggrecan [6,7,8], versican [9,10], perlecan [11], decorin, biglycan, fibromodulin, lumican, keratocan [6,12,13,14] and lubricin (PRG4) [15,16,17]. These have roles in weight-bearing, regulation of collagen fibrillogenesis and lubrication of the meniscal surface.

The β-defensins are 2–6 kDa cationic peptides which bind to meniscal proteoglycans and have anti-bacterial, ant-fungal and anti-viral activities which have protective roles to play in human meniscal tissues. β-Defensins are significantly upregulated during the inflammatory conditions which are generated in the knee-joint during OA and act in concert with DAMP (damage associated molecular pattern) and PAMP (pathogen associated molecular pattern) receptors such as the Toll-like and NOD-like receptors of the innate immune system which identify lipopolysaccharide and peptidoglycan components on cell membranes of invading organisms [19,20] and this response protects meniscal tissues from microbial invasion [21]. Aquaporin1 (AQP1) is a specific water transport channel protein expressed by articular chondrocytes, meniscal cells and synoviocytes in the knee-joint which is upregulated during OA and RA [21]. This may represent a new molecular target in therapeutic procedures aimed at maintaining the hydration and functional viscoelastic hydrodynamic properties of knee joint tissues during these arthritic conditions [22].

## 2. Studies on Meniscal Tissues in Health and Disease

### 2.1. Development of High Resolution Imaging and Sensitive Biomechanical Methodologies for the Analysis of Knee Joint Tissues

Meniscal biology in health and disease [2,3,4,5] and novel synovial fluid biomarkers associated with meniscal pathology have been identified [23] including synovial fluid cytokine profiles in chronic meniscal tears of the knee [24]. Attempts have been made to correlate meniscal damage with the pathological status of knee joint tissues using a number of imaging techniques including radiography [25], μCT imaging and histology [26], semiquantitative imaging of biomarkers of knee OA progression [27,28], synchrotron-based X-ray μCT imaging of biomarkers [29] and by comparison of molecular biomarkers with magnetic resonance (MR)-based cartilage composition and knee joint morphology [30]. Phase-contrast μCT is an emerging imaging technique, which is able to resolve 3D micro-scale structures in tissue samples without the need for staining or sectioning but also complements conventional histology providing additional information on the fine structure of meniscal tissues. 3D-mapping of the joint space for the diagnosis of knee OA using high resolution computed tomography has compared radiographic, normal and degenerate meniscal pathology [31]. Bioluminescence and second harmonic generation imaging reveal dynamic changes in the inflammatory and collagen landscape in early OA [32]. Ex vivo quantitative multiparametric magnetic resonance imaging (MRI) mapping has been used to assess human meniscus degeneration [33]. Ex vivo quantitative multiparametric MRI mapping uses 3.0-T MRI using inversion-recovery (T1), spin-lock multi-gradient-echo (T1ρ), multi-spin-echo (T2) and multi-gradient-echo (T2* and UTE-T2*) sequences to determine relaxation times of quantitative MRI (qMRI) parameters to provide information on the structure of normal and degenerate meniscal tissues. Biomechanical properties of the murine meniscus surface using atomic force microscopy (AFM)-based nano-indentation methodology has been used to establish topographical structure-function relationships in pathological versus normal menisci [34]. Nanoindentation AFM techniques offer high spatial resolution and force sensitivity for the measurement of the mechanical properties of biomaterials and tissues. Laboratory-based X-ray phase-contrast μCT imaging has been applied to the imaging of mouse articular cartilage in models of OA and is a promising research tool for the identification of new OA therapeutic targets at higher spatial resolution than possible with MRI techniques offering the possibility of the detection of early cartilage lesions which if untreated will develop into advanced OA lesions [35]. With the difficulty of obtaining time on synchrotron facilities for imaging, laboratory based X-ray μCT scanners are being developed to provide images of comparable quality to synchrotron images [35].

### 2.2. Identification of Biomarkers of Degenerate Meniscal Pathology

Several prospective biomarkers have been correlated with various criteria of meniscal pathology including synovial fluid HMGB (high mobility group box)-1 levels [36,37], inflammatory cytokines and biomarkers of cartilage metabolism 8 years after ACL (anterior cruciate ligament) reconstruction in operated and contralateral knees [38], synovial chemokine expression [34], synovial fluid MMP activity in torn menisci [39] and transcriptomic analysis of synovial extracellular RNA conducted following knee trauma [40]. These studies have suggested probable roles for specific biomolecular markers in meniscal degeneration. Meniscal cells are responsive to the inflammatory cytokines IL-1α and TNFα in an ovine in-vitro model where zonal analyses can also be undertaken to compare the phenotypic characteristics of inner and outer zone meniscal cells [41,42]. This demonstrated the involvement of meniscal pathology in OA development and its contribution to global joint disease [20]. Meniscus induced cartilaginous damage has also been shown to result in anatomical progression of early-stage OA in a canine model [43] emphasising the importance of meniscal pathobiology in global knee joint integrity [3]. Furthermore, targeted mutation of NOV/CCN3 in mice disrupts joint homeostasis and can induce OA-like symptoms [44]. On the other hand enhanced E-cadherin and PPAR expression leads to increased cell proliferation in primary human meniscal cells and these are potential molecular targets for meniscal regeneration [45].

Proteomics approaches have identified meniscal markers associated with disease pathology [46,47,48,49]. Genetics studies have demonstrated that Hajdu Cheney syndrome mice display an altered B-cell allocation during immune responses and are sensitized to the development of OA [50,51]. In contrast, the articular cartilage of Ctsk(−/−) mice is protected from the development of OA but has associated cellular and molecular changes in subchondral bone and cartilage ECM [52]. Elf3 contributes to cartilage degradation in-vivo in a surgical model of post-traumatic OA [53]. Transcriptomics has compared menisci from patients with and without OA [54]. Wnt7a inhibits interleukin-1 beta (IL-1β) Induced catabolic gene expression and prevents articular cartilage damage in experimental OA [55], Wnt16 also antagonises excessive canonical Wnt activation and protects cartilage in OA [56]. Furthermore, inhibition of Wnt/β-catenin signaling ameliorates OA in a murine model of experimental OA [57]. Deletion of Runx2 in articular chondrocytes decelerates the progression of medial meniscal destabilised induction of OA in adult mice [58]. Screening for characteristic genes in OA induced by destabilization of the medial meniscus utilizing a bioinformatics type approach has identified genes associated with meniscal pathology [59].

### 2.3. Gene Profiling and Transcriptomics of Meniscal Tissue and Articular Cartilage Provide Insights into the Etiopathogenesis of OA and Identify Potential Therapeutic Targets

Gene profiling has established that traumatic and degenerative meniscal tears have different expression profiles [60] and that gene expression in human meniscal tears has limited association with early degenerative changes in knee articular cartilage [61]. Comprehensive expression analysis of microRNAs and mRNAs in synovial tissue in a mouse model of early post-traumatic OA [62] and regional gene expression analysis of multiple tissues has been undertaken [63]. Gene profiling of articular cartilage from knees with meniscal tears are distinctly different from knees with OA [64]. The presence of meniscal tears also alters the gene expression profile evident in anterior cruciate ligament injury [65]. These studies are insightful as to different forms of OA which develop following meniscal or ligament injury [66]. The use of animal meniscal models [43,67,68,69,70] have also contributed significantly to a greater understanding of the pathobiology of meniscal degeneration and traumatic meniscal injury and may aid in the formulation of therapeutic interventions on these meniscal tissues [71,72].

The underlying molecular changes underlying progression of OA are incompletely understood however some transcriptomic studies are now providing some insightful information into this disease process. Gene profiling of articular cartilage from knees with meniscal tears are distinctly different from knees with OA cartilage lesions but relatively intact menisci [64], the presence of meniscal tears also alters gene expression in anterior cruciate ligament injury [65]. These studies are insightful as to different forms of OA which develop following meniscal or ligament injury [66] and stress the importance of examination of meniscal gene expression in destabilised menisci directly to obtain gene expression data of relevance to meniscal disease processes.

Gene expression studies in cartilage of surgically induced OA in wild-type (WT) mice and Adamts5Δcat mice, in which ADAMTS-5 activity is ablated and thus aggrecan loss and cartilage erosion is inhibited has been undertaken to distinguish gene expression changes that are independent of ADAMTS-5 activity and cartilage breakdown. Gene expression in meniscal cartilage from the developing lesion in the destabilized medial meniscus and corresponding regions in sham-operated joints were compared using whole-genome microarrays [73]. Previously identified OA-related genes, including Ptgs2, Crlf1, and Inhba, and novel genes, such as Phdla2 and Il11, were up-regulated in both WT mice and Adamts5Δcat mice, indicating that these are regulated independently of ADAMTS-5 activity. Other genes, such as Col10a1, a marker of cartilage hypertrophy and terminal differentiation, and genes of the Wnt/β-catenin pathway however were dependent on ADAMTS-5 activity. Foxo4, and Xbp1 endoplasmic reticulum-stress cell death transcriptional networks were activated by meniscal destabilisation. Many degradative protease genes, including Mmp3, Capn2, and the novel cartilage proteases Prss46 and Klk8 were also upregulated by meniscal destabilisation and the development of OA changes. Identification of genes that act independently of ADAMTS-5 identify these as genes which may initiate meniscal degeneration in OA and are prospective new therapeutic targets. Thus a complex picture of meniscal degradomics is emerging in OA.

Integrative epigenomics, transcriptomics and proteomics of patient chondrocytes has been used to identify cartilage genes and pathways involved in OA [74]. Examination of genes and transcription pathways that mark OA progression in isolated primary chondrocytes from paired intact versus degraded articular cartilage from knee-joint replacement donors has been undertaken. Genome-wide DNA methylation, RNA sequencing, and quantitative proteomics has been applied to these samples. Forty nine differentially regulated genes have been identified in intact versus degraded cartilage, 16 of these genes had not previously been implicated in OA progression. Three genes were particularly strongly identified across all analytic platforms used, namely *AQP1*, *COL1A1* and *CLEC3B* and an increased distribution of these proteins confirmed in OA cartilage using immunohistochemistry [74]. Integrated pathway analysis showed the involvement of ECM degradation, collagen catabolism and angiogenesis as important components of disease progression [74]. The growth of blood vessels and nerves are closely linked processes that share common regulatory mechanisms [75]. Histology of OA and normal cartilage has also demonstrated greater blood vessel ingrowth in tissues with more advanced OA [74].

As already noted, three genes have been identified strongly associated with OA cartilage, aquaporin-1 (*AQP1*), tetranectin (*CLEC3B*) *and COL1A1* [74]. AQP1 facilitates water transport across biological membranes. In OA cartilage increased cartilage hydration leads to chondrocyte swelling which has been suggested to contribute to OA development [76]. AQP1 is also overexpressed in the meniscus in a rat model of OA [22], and its levels are elevated in degenerate human articular cartilage specimens compared to normal intact cartilage [77]. Tetranectin has also been previously shown to be upregulated in OA [78,79]. Collagens are the main fibrillar structural components of cartilage, disruption of collagen biosynthesis and elevated degradation of type-I collagen by MMPs have important roles in the progression of OA [80,81]. A search of Drugbank [82] for existing drugs approved for human use for the treatment of OA, identified Acetazolamide, a carbonic anhydrase inhibitor diuretic agent which inhibits AQP1 and Tenecteplase, a thromobolytic agent which binds tetranectin [74] as candidates for further evaluation. However it remains to be established how effective these compounds will be as therapeutic agents targeting the degenerate meniscus before any assessment can be made as to whether they will have a global impact on the knee-joint as a therapeutic agents for the treatment of OA.

## 3. Current Approaches in Meniscal Repair

The potential of mesenchymal stem cells (MSCs) for tissue repair became evident in 2003 [83] when adult stem cells isolated from bone marrow were used in a caprine model where the medial meniscus was completely removed and the anterior cruciate ligament (ACL) was excised. Marked regeneration of the medial meniscus was evident in MSC treated joints, and degeneration of the articular cartilage, osteophytic remodeling, and subchondral sclerosis were reduced compared to joints treated with vehicle alone without cells, however there was no evidence of repair of the ACL in any of the joints. This study clearly showed that local delivery of adult MSCs to injured joints stimulated regeneration of meniscal tissue and retarded the progressive destruction of other tissue in the knee joint which is normally seen in animal meniscectomy models of OA [84,85,86,87,88].

Many reviews have emphasized the great potential of MSCs for meniscal repair applications [25,31,36,40,42,43,45,71]. MSCs have been employed in numerous strategies aimed to promote repair of meniscal lacerations [23] and defects in the avascular inner meniscal zone [39,89,90,91] or in meniscal replacement in meniscectomised animals [83]. A number of animal models have been employed in these studies including goats [83], rabbits [92,93], minipigs [69,94], sheep [84,85,86,87,88,95] and rats [26,29,63,96]. MSCs have been sourced from bone marrow [95,97,98,99,100,101], adipose tissues [98,99,102,103] and synovium [62,94,98,103]. The utility of a number of bioscaffolds seeded with therapeutic cells have also been examined to promote repair of inner zone meniscal defects (reviewed in [18]). These scaffolds include a stem cell/collagen-derived scaffold [91], polyurethane [104] and a scaffold-free tissue-engineered construct derived from allogeneic synovial MSCs [98] besides a number of conventional tissue engineering synthetic resorbale bioscaffolds [105]. Fibrin-glue has also been evaluated as a cell delivery vehicle for meniscal repair [89,97,103,106]. Examination of the cell directive properties of MSCs over meniscal cells co-cultured in scaffold free micromass pellet cultures has demonstrated a significant increase in cellular proliferation, type I and II collagen and aggrecan production [105]. Three-dimensional coculture of meniscal cells and MSCs in collagen Type I hydrogels have also demonstrated positive responses in terms of matrix production conducive to meniscal repair processes [107]. Evaluation of effects on cell shape, matrix production, and mechanical performance of neo-tissues produced by such 3D MSC: meniscal cell co-cultures has shown positive effects of potential application in the repair of meniscal defects [31,108]. Direct cell-cell contact between MSCs and meniscal cells is not required to elicit such positive responses since MSC conditioned media can elicit similar responses in meniscal cells due to trophic factors transferred from the MSCs by paracrine effects. Transfer of material from MSCs can also be undertaken by microvesicles (exosomes) and these are of clinical potential as a means of directing positive therapeutic responses in resident meniscal cell populations [109,110,111].

The meniscus is a tissue which has very significant physical demands placed on it within the knee joint to protect and stabilise other knee joint tissues which provide knee joint articulation and weight bearing. Artificial scaffolds developed for partial or total meniscal replacement still do not totally reproduce the material properties of native meniscal tissues and further developments will be required to perfect this technology [112,113,114]. Stereolithography [115] (3D printing) of individualised menisci or meniscal segments with bioscaffolds which can be seeded with autologous stem cells [108] to eventually synthesize a replacement meniscal structure represents a promising advancement in this area of meniscal repair.

Knowledge of biological glues and tissue adhesives for the attachment of meniscal flaps or tears have undergone significant improvements in recent years. Tissue adhesives currently used in surgical applications, reviewed in [18,105] include the topical skin adhesive Dermabond^®^ (2-octyl cyanoacrylate), fibrin glue for pulmonary leaks, and the recently developed TissuGlu^®^ (an FDA approved urethane-based adhesive) for abdominoplasty surgery [116,117]. Tissue adhesives inspired by high performance adhesive marine compounds which firmly attach mussels [118] and barnacles [119] to marine substrates are promising tissue adhesives. An interesting biological glue produced by the Australian native frog species *Notaden benetii* (also known as the crucifix toad or holycross frog) is a skin exudate secreted by the female frog during mating and ensures extended sexual union by the male for efficient egg fertilization. This frog glue [120,121] provides superior adhesive properties to fibrin glue, is non-immunogenic and has been successfully trialled in the re-attachment of a bucket handle tear and in rotator cuff repair surgery, a re-engineered development of this promising polymer deserves to be undertaken [122,123]. This colourful digging toad occurs in the semi-arid grasslands and black soil plains of central inland New South Wales and the interior of southern Queensland, west of the Great Dividing Range in Australia. The use of bioadhesive or hydrogels as delivery systems for therapeutic cells, drugs or antibiotics is another innovative development in repair biology but has yet to be applied in meniscal repair procedures [124,125,126,127,128].

Moderate physical exercise has been examined as a means of positively regulating beneficial chondroprotective and anti-inflammatory genes in human knee joint tissues as a means of therapeutic intervention during inflammatory osteoarthritic conditions in the knee-joint [129]. Such an approach has delivered some beneficial effects in type B synoviocytes which suggests that moderate exercise may delay the onset of degenerative changes in knee-joint tissues in OA and RA [130]. Moderate physical exercise has also been shown to improve the lubrication of articular cartilage and joint function in ageing rats through an upregulation in lubricin expression [129]. Excessive joint loading also down-regulates lubricin expression and decreases the functional properties of knee-joint tissues in an ovine meniscectomy model [131,132]. A virgin-olive oil supplemented diet in combination with moderate physical exercise has been shown to ameliorate degenerative effects on knee-joint tissues in an ACL transection model of OA in rats [133]. This treatment upregulates lubricin expression which has a beneficial effect on knee-joint function, a recent study has suggested that lubricin can regulate knee-joint inflammation. These findings therefore suggest that this simple dietary modification may be a useful medical therapy to use in combination with moderate exercise in man to prevent osteoarthritic disease progression not only preserving the articular cartilage and improving its lubricative properties and thus joint articulation but may also be globally beneficial to all knee-joint tissues [133].

## 4. Conclusions

The menisci are major cartilaginous structures in the human knee with essential roles in the stabilization and weight bearing of this structure. A healthy meniscus is therefore a high priority to ensure the health of other knee joint tissues. A realization of the inter-dependance of all knee joint tissues to provide an efficient articulatory and weight bearing structure indicates that for any therapeutic measure to meet with success in the treatment of OA or RA it must be beneficial to all of the varied cell populations in the human knee. The large number of therapeutic measures being developed to specifically halt further degeneration, stimulate regeneration or replace meniscal tissues points to the central importance of the meniscus to knee joint function. Without a therapeutic measure effectively addressing all these parameters in OA there is little likelihood of a successful outcome in this most debilitating of conditions. Moderate exercise has been shown to be beneficial to knee joint tissues and dietary adjustment to include virgin olive oil has further beneficial effects in terms of a lowering of knee joint inflammation. With knee OA predicted to become a leading musculoskeletal condition within the next two decades there is an extreme need for effective measures to be developed and all avenues of therapy need careful evaluation.

## Figures and Tables

**Figure 1 cells-08-00324-f001:**
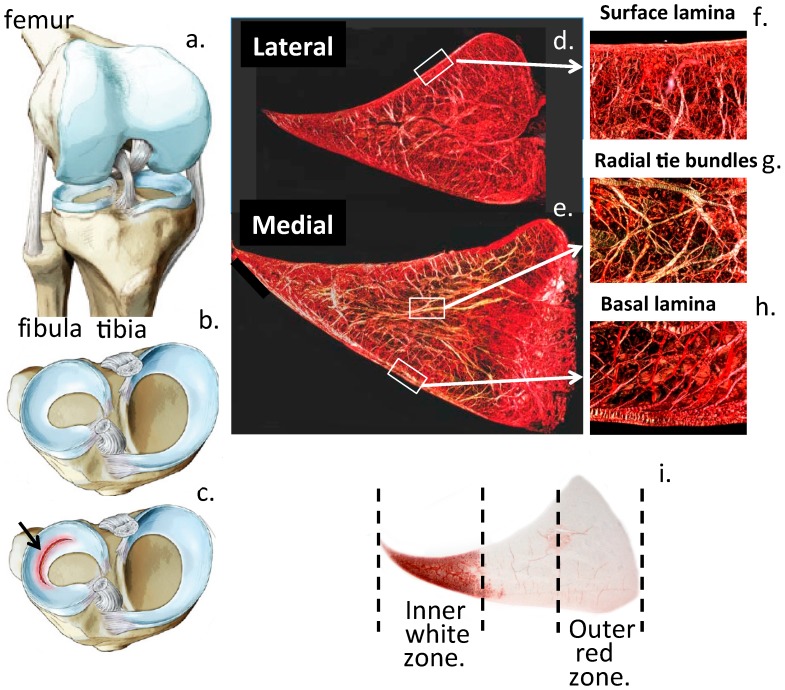
Knee anatomy showing the location of the knee joint menisci (**a**, **b**) with a radial tear arrowed (**c**) and the collagen fibre organisation of the menisci in vertical sections through lateral (**d**) and medial menisci (**e**) which equip these as weight bearing stabilising structures in the knee joint. Radial tie bundles and collagen fibrillar arrangements in the surface and basal meniscal laminas are also shown (**f**–**h**). The inner zone meniscal cells are cartilaginous whereas the outer zone meniscal cells are more fibrocartilaginous (**i**). This is clearly evident in the focal localisation of the HS-proteoglycan perlecan which is a chondrogenic marker [10]. Figure modified from [18] with permission of InTech Open Publishers: London. Figure copyright J. Melrose.

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
