# Peer review of "The Importance of the Knee Joint Meniscal Fibrocartilages as Stabilizing Weight Bearing Structures Providing Global Protection to Human Knee-Joint Tissues"

_cells, 2019, doi:10.3390/cells8040324_

Round 1

Reviewer 1 Report

This manuscript characterises and summerazies the data about menisci replacement and pathologycal conditions of the fibrocartilage. The manuscript is logically organized, but in some points it is not discussed deeply enough for a review article.

 Minor suggestions:

6. Animal models for the evaluation of meniscal repair using bioscaffolds is more related to bioengenering

Page 1 L39 "articulation [2] "c

Page 2 L46 " lumican, ", extra space

Page 2 L72 "in-vitro" hyphen is not needed

 Page 4 L135 "marrow[78, 80-84]," , space is missing

Page 4 L153 "populations[93-95]." , space is missing

Page 5 L159 "[99](3D printing)" , space is missing

Author Response

I HAVE ATTACHED MY COMMENTS AS A PDF FILE

Reviewer 2 Report

First of all, I would like to thank you for the opportunity to review your work, and for the contribution that you are making to the field. 

The major concerns I have with the paper concern sections 2 and 4. In section 2, you discuss several studies of meniscus health/damage using various visualization methods, but often don't summarize the major findings. For instance, in the sentences, "Ex vivo quantitative  multiparametric MRI mapping has been used to assess human meniscus degeneration. Biomechanical properties of the murine meniscus surface using AFM-based nano-indentation methodology has been used to establish topographical structure-function relationships in pathological versus normal menisci," I am left with the understanding that these studies are important, but I do not understand what we learned from them. 

In section 4 you discuss several studies that have examined differences in gene expression between healthy and damaged meniscal tissue. You note that studies found differences between these two meniscus states, but don't discuss the genes involved. Are the same genes implicated in section 3 also prominent in the results of section 4? Are there particular signaling pathways that relate to meniscus health. or an overall model of how genetic regulation of meniscus health/repair functions? Any clarification you can being to this would be a huge help to novice readers. 

Apart from that, I would recommend moving the last paragraph from section 7 to the beginning of the introduction, as it provides needed context for the purpose of your review and helps ease the reader into your discussion of the topic. 

Finally, I think that the paper would be easier to approach if it was organized into sections and subsections, with sections 2-4 grouped under a section (e.g. Gene expression in meniscal health and damage) and sections 5-7 group together (something like Current methods in meniscus repair)

Apart from that, I thought that your article makes a useful contribution to the field and that your discussion for the current and emerging meniscus repair methods was particularly clear. 

Author Response

(The authors gave the same response as above.)

Reviewer 3 Report

The manuscript entitled “The Importance of the knee joint meniscal fibrocartilages as stabilizing weight-bearing structures providing global protection to human knee-joint tissues” deals an important issue of medical fibrocartilage biology. 

Manuscript’s content is interesting and well written, the scope is sufficient and concisely expressed, the format is appropriate. The review is innovative and could be a proof of concept in this field of research. Moreover, the paragraphs are very representative to demonstrate and to strengthen the current literature results. However, there are some minor concerns that should be resolved before recommending publication. 

Please improve the aim of this study in the abstract and in the introduction section to help better readers understanding.

In the introduction please add some important and fundamental details regarding the AQP1 and β-defensin-4 biomarkers related to menisci. I recommend checking the following interesting and fundamental papers or others and comment them in relation to the study topic: 

Expression of β-defensin-4 in "an in vivo and ex vivo model" of human osteoarthritic knee meniscus.

Aquaporin 1 (AQP1) expression in experimentally induced osteoarthritic knee menisci: an in vivo and in vitro study.

Pag 5, lines 181-192 please add adequate references as follow, talking also about the non-pharmacologic treatment such as physical activity and nutrition. 

Moderate physical activity as a prevention method for knee osteoarthritis and the role of synoviocytes as biological key. IJMS 2019, 20(3), 511.

Physical activity ameliorates cartilage degeneration in a rat model of aging: a study on lubricin expression. Scand J Med Sci Sports. 2015 Apr;25(2):e222-30.

Extra-virgin olive oil diet and mild physical activity prevent cartilage degeneration in an osteoarthritis model: an in vivo and in vitro study on lubricin expression. J Nutr Biochem. 2013 Dec;24(12):2064-75.

Please strengthen and improve the conclusion, adding the clinical relevance of your work and some important suggestions for the scientific community. 

Please refresh and update the reference list section. 

Figure 1 is very representative, congrats. 

Author Response

MY RESPONSES ARE ATTACHED AS A PDF FILE
